# Valve turning towards on-cycle in cobalt-catalyzed Negishi-type cross-coupling

Xu Luo[1,9], Dali Yang[1,9], Xiaoqian He[2,9], Shengchun Wang [1], Dongchao Zhang[1], Jiaxin Xu[1], Chih-Wen Pao[3], Jeng-Lung Chen [3], Jyh-Fu Lee[3], Hengjiang Cong [1], Yu Lan [2], Hesham Alhumade [4,5], Janine Cossy [6] ✉, Ruopeng Bai [2] ✉, Yi-Hung Chen [1] ✉, Hong Yi [1,7] ✉ & Aiwen Lei [1,8] ✉

Ligands and additives are often utilized to stabilize low-valent catalytic metal species experimentally, while their role in suppressing metal deposition has been less studied. Herein, an on-cycle mechanism is reported for $CoCl_2bpy_2$ catalyzed Negishi-type cross-coupling. A full catalytic cycle of this kind of reaction was elucidated by multiple spectroscopic studies. The solvent and ligand were found to be essential for the generation of catalytic active Co(I) species, among which acetonitrile and bipyridine ligand are resistant to the disproportionation events of Co(I). Investigations, based on Quick-X-Ray Absorption Fine Structure (Q-XAFS) spectroscopy, Electron Paramagnetic Resonance (EPR), IR allied with DFT calculations, allow comprehensive mechanistic insights that establish the structural information of the catalytic active cobalt species along with the whole catalytic Co(I)/Co(III) cycle. Moreover, the acetonitrile and bipyridine system can be further extended to the acylation, allylation, and benzylation of aryl zinc reagents, which present a broad substrate scope with a catalytic amount of Co salt. Overall, this work provides a basic mechanistic perspective for designing cobalt-catalyzed cross-coupling reactions.

Cross-coupling reactions have been widely used to build carbon–carbon bonds to produce pharmaceuticals and materials since the 1970s using palladium catalyst[1,2]. The elucidation of the fundamental steps in the catalytic cycle is crucial, as it helps to understand the catalyst activity, stability, and deactivation. For instance, "metal deposition"[3] is one of the most common pathways for decomposing a homogeneous catalyst (Fig. 1A). The discovery of oxidatively stable ligands and coordinating solvents can induce a high catalytic turnover

of Pd(0)/Pd(II) cycle[4,5]. And the lifetime of the catalyst can be enhanced by choosing the right solvent[6].

Recently, the emergence of the development of earth-abundant 3d metal catalysts for cross-couplings has been increasing[7–13]. Although a Co-catalyzed Negishi-type cross-coupling reaction has been previously reported, the reaction mechanism is still unclear. Moreover, the conditions used for such cross-coupling are limited to highly active electron-deficient electrophiles[14–19]. Mechanistic studies are necessary

[1]College of Chemistry and Molecular Sciences, the Institute for Advanced Studies (IAS), Wuhan University, Wuhan 430072, P.R. China. [2]School of Chemistry and Chemical Engineering, Chongqing Key Laboratory of Theoretical and Computational Chemistry, Chongqing University, Chongqing 400030, P.R. China. [3]National Synchrotron Radiation Research Center, Hsinchu 30076, Taiwan. [4]K. A. CARE Energy Research and Innovation Center, King Abdulaziz University, Jeddah 21589, Saudi Arabia. [5]Department of Chemical and Materials Engineering, Faculty of Engineering, Center of Research Excellence in Renewable Energy and Power Systems, King Abdulaziz University, Jeddah 21589, Saudi Arabia. [6]Molecular, Macromolecular Chemistry, and Materials, ESPCI Paris, CNRS, PSL University, 75005 Paris, France. [7]Wuhan University Shenzhen Research Institute, 518057 Shenzhen, China. [8]State Key Laboratory of Organometallic Chemistry, Shanghai Institute of Organic Chemistry, Chinese Academy of Sciences, Shanghai 200032, P.R. China. [9]These authors contributed equally: Xu Luo, Dali Yang, Xiaoqian He. ✉e-mail: janine.cossy@espci.fr; ruopeng@cqu.edu.cn; yihungchen@whu.edu.cn; hong.yi@whu.edu.cn; aiwenlei@whu.edu.cn

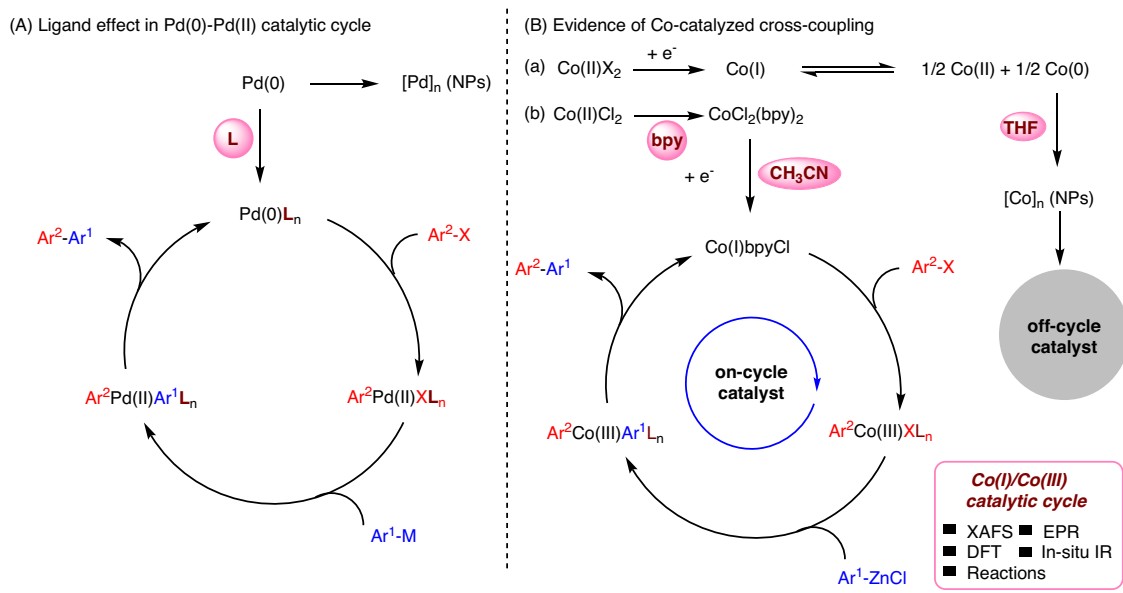

**Fig. 1 | Elucidation of the catalytic cycle. A** Pd-catalyzed cross-coupling. **B** Co-catalyzed Negishi-type cross-coupling.

to optimize the current reaction conditions and extend the substrate scope. It was reported that the Co(I) species could undergo a disproportionation leading to solid cobalt along with Co(II) on the timescale of slow cyclic voltammetry (CV) (Fig. 1B-a)[20,21]. Due to the aggregation of Co(0), the concentration of catalytic species available for oxidative addition is decreased, and the reaction is off-cycle. Understanding the stability and reactivity of Co(I) could benefit the optimization of the reaction conditions. Solvent effects play an important role in the reactivity of the reaction rates, catalyst structure, and product selectivity[22]. They are applied in tunneling reactions[23]. Coordination power between solvents and metal ions is related to the solvation Gibbs free energy[24]. As there is a consensus that ligand backbone can control the comproportionation and disproportionation of Ni species[25], we speculate that ligand effects can maintain the propagating on-cycle cobalt species.

Herein, different reductive valences of cobalt species have been investigated in THF and CH₃CN solvents. The stability form of Co(I) complex is regulated by acetonitrile and bipyridine, which is conducive to the on-cycle catalytic pathway. While disproportionation of Co(I) to Co(0) is prone to the off-cycle pathway in THF (Fig. 1B-a). The detailed mechanistic investigations of Co-catalyzed Negishi-type coupling (Csp²–Csp²) in acetonitrile are reported by X-ray absorption fine structure spectroscopy (XAFS), electron paramagnetic resonance (EPR), IR, density functional theory (DFT) calculations, and synthesis. The spectroscopic studies provide reliable evidence for the DFT calculation of the structural simulation. DFT calculations help us to understand the experimental results and provide guidance to determine the intermediates of the catalytic cycle. By using these techniques, a Co(I)/Co(III) catalytic cycle, including structural information, was elucidated. The resulting Co(I) species are stabilized by electrophile, ligand, and solvent, preventing disproportionation. The mechanistic studies guided us to establish the reaction conditions with a comprehensive substrate scope that was further extended to acetylation, allylation, and benzylation of aryl zinc derivatives.

## Results and discussion
XAFS spectroscopy is frequently employed in heterogeneous catalysis to determine electronic and local geometric structures in a specific element. On the contrary, XAFS spectroscopy is less often used for the characterization of organometallic reagents and the characteristic of homogeneous catalysts[26–29]. However, recently time-resolved XAFS by

a custom-made flow cell has been used by Bedford et al.[30]. to study the iron-catalyzed Negishi cross-coupling, and XAFS spectroscopy can be a precious tool to study transition metal-catalyzed reactions[31–33].

## Mechanistic studies in THF
A range of cobalt-catalyzed Kumada cross-couplings has been reported involving cobalt complexes of chelating amines or phosphines, including mechanistic studies[34–41]. Initially, we want to explore whether cobalt-catalyzed Negishi cross-couplings go through the same path. However, no cross-coupling product **3aa** was detected in tetrahydrofuran (THF, Fig. 2A-a), which could come from the reduction of cobalt disproportionation to inactive low-valent cobalt, which renders the oxidative addition. Therefore, the X-ray absorption near-edge structure (XANES) spectrum was used to identify the valence of cobalt after the addition of PhZnCl. The reduction of CoCl₂ with PhZnCl in THF was found to be Co(0) on account of the Co-Co bond formation (Fig. 2A-c). Co(0) may be prone to "metal deposition" as Pd, leading to unactive nanoparticles. This could explain why cobalt cannot catalyze cross-coupling reactions in THF.

## Mechanistic studies in CH₃CN
Thus, the influence of different solvents on the reaction was investigated (Supplementary Table 1) delightfully, in CH₃CN, the cross-coupling product **3aa** was formed in 62% isolated yield. In this solvent, the disproportionation of Co(I) seems to be suppressed, leading to a sufficient concentration of active Co(I) species. To verify this hypothesis, CoCl₂ with PhZnCl (**2a**, 10 equiv.) in CH₃CN was carried out, monitored by Quick-XAFS (QXAFS) on the timescale of 2 scans per second[28] (Supplementary Fig. 3a). In the absence of an electrophile, the Co(I) species underwent a disproportionation leading to Co(0) and cobalt nanoparticles upon increasing temperature. However, in the presence of 4-bromobenzonitrile (**1a**) at room temperature (Supplementary Fig. 3c), Co(I) species can undergo an on-cycle catalytic pathway rather than disproportionation. Although 4-bromobenzonitrile (**1a**) undergoes a smooth cross-coupling with arylzinc reagent **2a** without a ligand, the electron-rich electrophile **1b** demonstrated that the presence of a ligand is important (Fig. 2B-a). In situ IR showed that the initial reaction rate increased along with the increasing concentration of **1a** and CoCl₂ in the absence of a ligand (Supplementary Fig. 8). This indicates that oxidative addition may be the rate-determine step of the

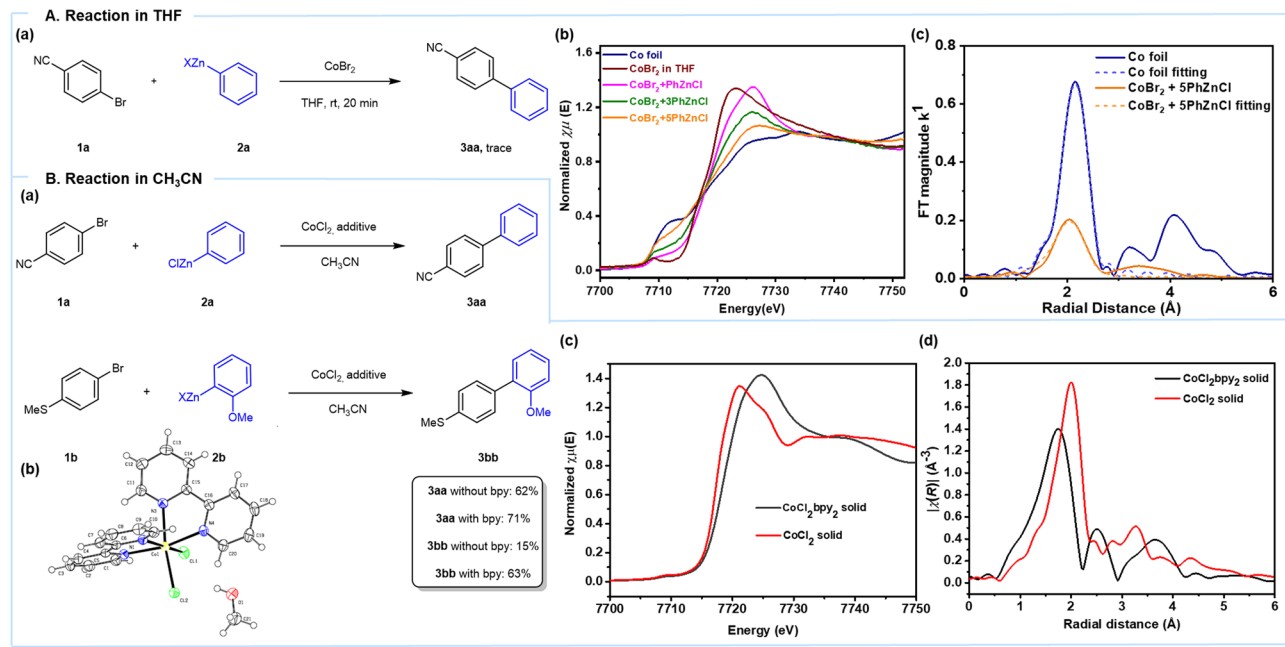

**Fig. 2 | Reactions in THF and CH₃CN. A** Reaction in THF and mechanistic studies by XAFS. a Reaction in THF; b XANES data of the reaction between PhZnCl and CoBr₂; c Fourier Transform (FT) Magnitude of Co(0) species in the reaction and Co foil. **B** Reaction in CH₃CN and characteristic data. a Reaction in CH₃CN with and without bpy; b single-crystal X-ray diffraction of CoCl₂(bpy)₂, crystal growth from MeOH; c XANES data of CoCl₂(bpy)₂ and CoCl₂; d FT-XAFS data of CoCl₂(bpy)₂ and CoCl₂.

reaction without ligand, and decreasing yield without bpy ligand when electron-rich electrophile is used.

To gain an in-depth understanding of the ligand and solvent effect on the Co-catalyzed Negishi-type cross-coupling, the catalyst structure and oxidation state in the catalytic cycle were further elucidated. The structure of the pre-catalyst [CoCl₂(bpy)₂], in an octahedral geometry, was confirmed by single-crystal X-ray diffraction (Fig. 2B-b, CCDC number is 2040561). As shown in Fig. 2B-c, XANES data were recorded for the pre-catalyst complex and the commercially available CoCl₂ (99.7%). The edge energy of [CoCl₂(bpy)₂] (7718.4 eV) is higher than CoCl₂ (7717.4 eV) because of the different coordination and spatial configuration. In the case of CoCl₂, the EXAFS signal is dominated by Co−Cl contributions. A relatively short bond length in CoCl₂(bpy)₂ is characteristic of Co-N and Co-Cl average bond length (Fig. 2B-d). The fitting data of the EXAFS spectrum of CoCl₂(bpy)₂ has been performed using the Artemis program (Supplementary Table 3). The cobalt atom is coordinated by four nitrogen and two chloride atoms in the first shell, and the bond lengths of Co-N and Co-Cl are 2.169 Å and 2.419 Å, respectively. In the second shell, the Co-C bond can be deduced with a length of 3.043 Å. These results were consistent with the data of the single-crystal structure.

The identification of the active species through various experiments was continued. The edge energy of the pre-catalyst CoCl₂(bpy)₂ is located at 7718.4 eV and shifts after the addition of PhZnCl [CoCl₂(bpy)₂:PhZnCl = 1:10] to 7716.5 eV, which might correspond to the reduction of Co(II) to Co(I) (Fig. 3a, red to blue), moreover, it was further supported by EPR spectroscopy (Fig. 3h). In addition, it was confirmed that 4-bromobenzonitrile (1a) did not coordinate with the pre-catalyst CoCl₂(bpy)₂ (Supplementary Fig. 4). CoCl₂(bpy)₂ was then treated with PhZnCl in the presence of 1a to investigate the whole catalytic cycle. The oxidative addition of 1a is a two-electron process, which may lead higher valence state of Co species (Supplementary Fig. 5). The low Co EPR signal can be rationalized by forming EPR-silent Co(I) or Co(III) species which could be generated at room temperature. (Supplementary Fig. 2).

To get solid evidence of Co(I) species, EXAFS fitting and ORCA pre-edge calculation of [CoCl₂(bpy)₂] and the addition of PhZnCl

[CoCl₂(bpy)₂:PhZnCl = 1:10] were achieved under 0 °C. The magnitude and imaginary of [CoCl₂(bpy)₂ + 10 PhZnCl] provide evidence that PhZnCl reduced CoCl₂(bpy)₂ to Co(I) species. In the course of activation, the fitting results of EXAFS data indicate that the Co(I) species 1 is coordinated by two nitrogen atoms (bond length = 2.070 Å) and one chloride atom (bond length = 2.295 Å) (Fig. 3c–f and Supplementary Table 4). The XANES of Co(I) (blue line) in Fig. 3a, which has an obvious pre-edge and broad main post-edge feature compared to Co(II), could possibly be due to the short Co−N & Co−Cl bonds in a 3-coordinated Co(I) and the back-bonding character between bpy and Co. In addition, ORCA pre-edge TDDFT calculation showed that the pre-edge intensity of Co(I) species 1 matches with the generated 3 coordinated Co(I) experimental data well, which is more intense than the pre-edge intensity of [CoCl₂(bpy)₂] catalyst as Co(II) species (Fig. 3g).

To rationalize the catalytic pathway, a series of experiments were conducted. Electron-deficient bromide and electron-rich iodide were chosen as competitive substrates for cross-coupling[42]. It could be mentioned that 4-bromobenzonitrile (1a) has a lower reductive potential than the 4-iodo-N,N-dimethylaniline (1d) (−2.47 V, −2.04 V for 1a; −2.57 V for 1d versus Ag/AgCl in CH₃CN, Fig. 4Ab). And when aryl-zinc derivative 2b reacted with a solution of 1d and 1a in the presence of CoCl₂(bpy)₂, the coupling product 3db, derived from 3ab, is dominant (Fig. 4Aa). Meanwhile, when vinyl-containing aryl iodide 1c[43,44] was subjected to cross-coupling using CoCl₂(bpy)₂, the biphenyl product 3cc was obtained instead of the cyclized product (Fig. 4Ac). This result suggested that a two-electron-based oxidative addition pathway is taking place instead of a single-electron pathway.

The desired cross-coupling products were obtained by premixing the electrophile and the catalyst, which should be followed by the zinc reagent addition at rt. However, a low yield was observed by premixing CoCl₂(bpy)₂ and PhZnCl in CH₃CN for 5 min, followed by the addition of the electrophile 1a (Fig. 4Ad). Nevertheless, a moderate yield was observed by premixing CoCl₂(bpy)₂ and PhZnCl in CH₃CN at 0 °C. Then electrophile 1a was added at 0 °C for 5 min and stirred at rt. We reasoned that the active Co(I) species were generated under standard conditions and could be stabilized at low temperatures. However, Co(I) complex leads to solid Co(s) and Co(II) by disproportionation in

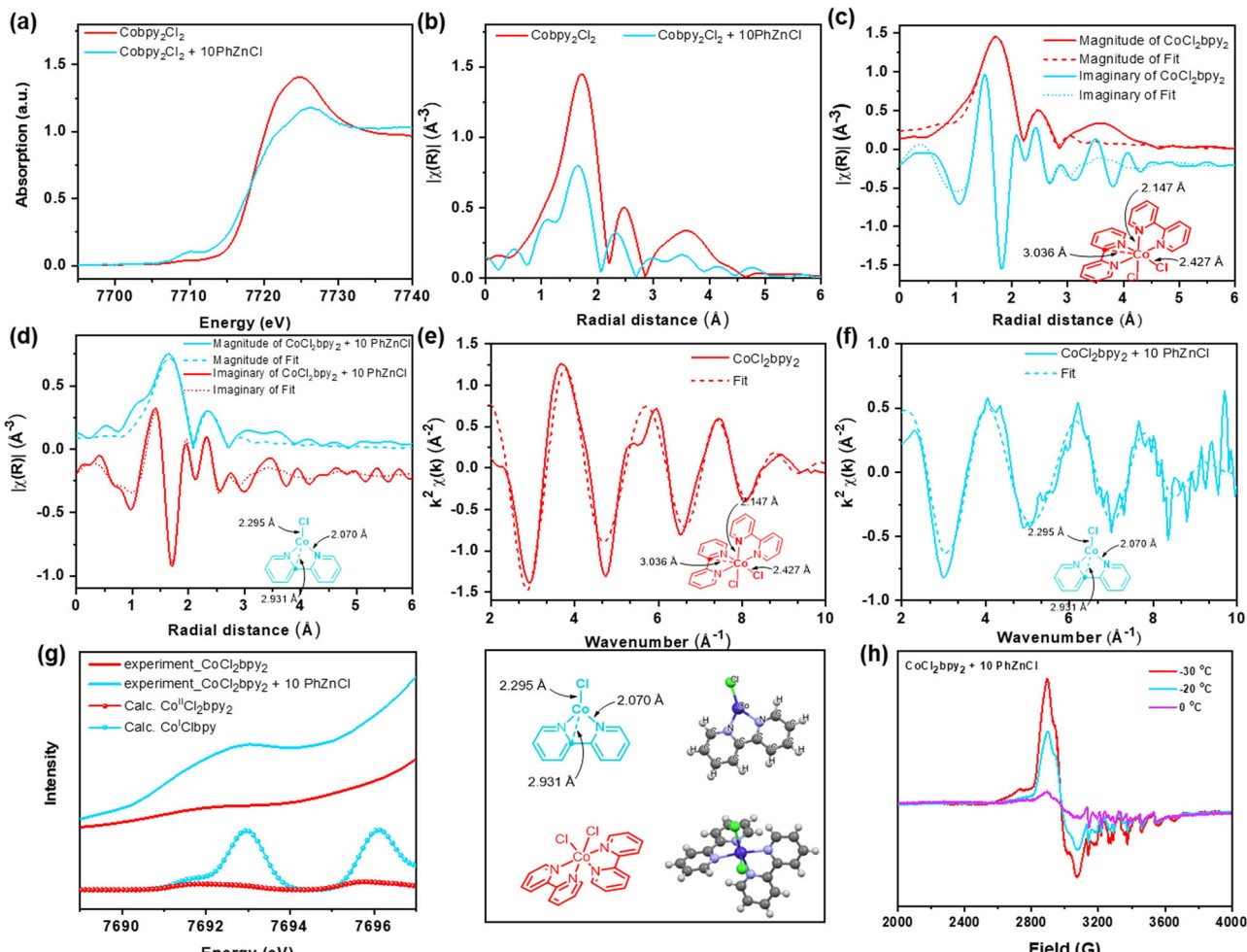

**Fig. 3 | Mechanistic studies by XAFS, ORCA calculation, and EPR spectroscopy.** **a** Normalized Co K-edge XANES spectra of CoCl₂bpy₂ in CH₃CN; CoCl₂bpy₂, and PhZnCl (10 equiv.) in CH₃CN. **b** FT-XAFS data of CoCl₂bpy₂; CoCl₂bpy₂ and PhZnCl (10 equiv.). **c** EXAFS Fit of CoCl₂bpy₂; **d** EXAFS Fit of CoCl₂bpy₂ and PhZnCl (10 equiv.). **e** EXAFS Fit of CoCl₂bpy₂ in K-edge. **f** CoCl₂bpy₂ and PhZnCl (10 equiv.) in K-edge. **g** ORCA pre-edge Calculation. **h** EPR signal of Co(II).

the absence of electrophile at rt, which results in low catalytic efficiency. Moreover, when 2,5-dibromothiophene reacted with *p*-fluorophenylzinc reagent under the optimized conditions, followed by iodolysis, 2-bromo-5-iodo-thiophene was formed in 12% GC yield (see SI, Scheme S1). This result indicates that (5-bromothiophen-2-yl) cobalt(III) species is reduced in solution, and (5-bromothiophen-2-yl) zinc(II) chloride might be generated, which could be identified by iodolysis.

## DFT calculations

To further investigate the mechanism for this Co(I)-catalyzed cross-coupling reaction, a detailed computational study was performed in Fig. 4B. Natural triplet cobalt(I) complex ³**1** was selected as the starting point for the free energy profile. Coordination of aryl moiety of 4-bromobenzonitrile **1a** to the cobalt center generates the triplet intermediate ³**10**, which is exergonic by 4.1 kcal/mol. Subsequent oxidative addition via transition state ³**TS-1** generates five coordinated aryl-cobalt(III) triplet intermediate ³**11**. The energy barrier is 8.6 kcal/mol, and this step is 2.6 kcal/mol exoergic. Transmetalation of bipyridyl coordinated phenylzinc chloride **2a'** to cobalt center via transition state ³**TS-2** generates diaryl-cobalt(III) intermediate ³**13** with an energy barrier of 21.8 kcal/mol. When using phenylzinc chloride **2a**, the energy barrier is 23.2 kcal/mol via transition state ³**TS-4**. This indicates that bipyridyl promotes a transmetalation process. Finally, direct reductive elimination of ³**13** via transition state ³**TS-3** occurs to release the

desired biarene product **3aa** and regenerates Co(I) species ³**1**, which energy barrier is 6.2 kcal/mol. Once natural diarylcobalt(III) intermediate ³**13** is generated, ligand exchange of chloride with bipyridyl could generate unstable triplet Co(III)-intermediate ³**15**. Subsequently, the energy barrier of the reductive elimination, via transition state ³**TS-5**, is 12.2 kcal/mol, and 6.0 kcal/mol higher than natural reductive elimination transition state ³**TS-3**. Therefore, this pathway is dynamically unfavorable. The overall activation energy of the reaction is 21.8 kcal/mol, and the transmetalation is determined to be the rate-determining step. We also consider the singlet Co(I)-catalyzed cross-coupling reaction, computational studies show that the singlet free energy profile has a higher energy barrier than the triplet free energy profile. Therefore, this pathway was ruled out (see the SI for details)[45,46].

Moreover, the kinetic relationship of each component was tested under in situ IR with **1a**, PhZnCl, and [CoCl₂(bpy)₂]. The initial reaction rate is nearly independent of the concentration of electrophile **1a** (Fig. 5A-a). Therefore, the reaction can be compatible with electron-rich electrophiles with bpy as the ligand. This is consistent with the DFT calculation. The initial reaction rate is positively correlated with the concentration of [CoCl₂(bpy)] catalyst (Fig. 5A-b). However, when we tried to change the concentration of PhZnCl, the kinetic relationship was not obvious. It is because zinc reagent is not only involved in the reduction process of cobalt catalyst but also in the process of transmetalation, reductive elimination, so the pattern is not obvious.

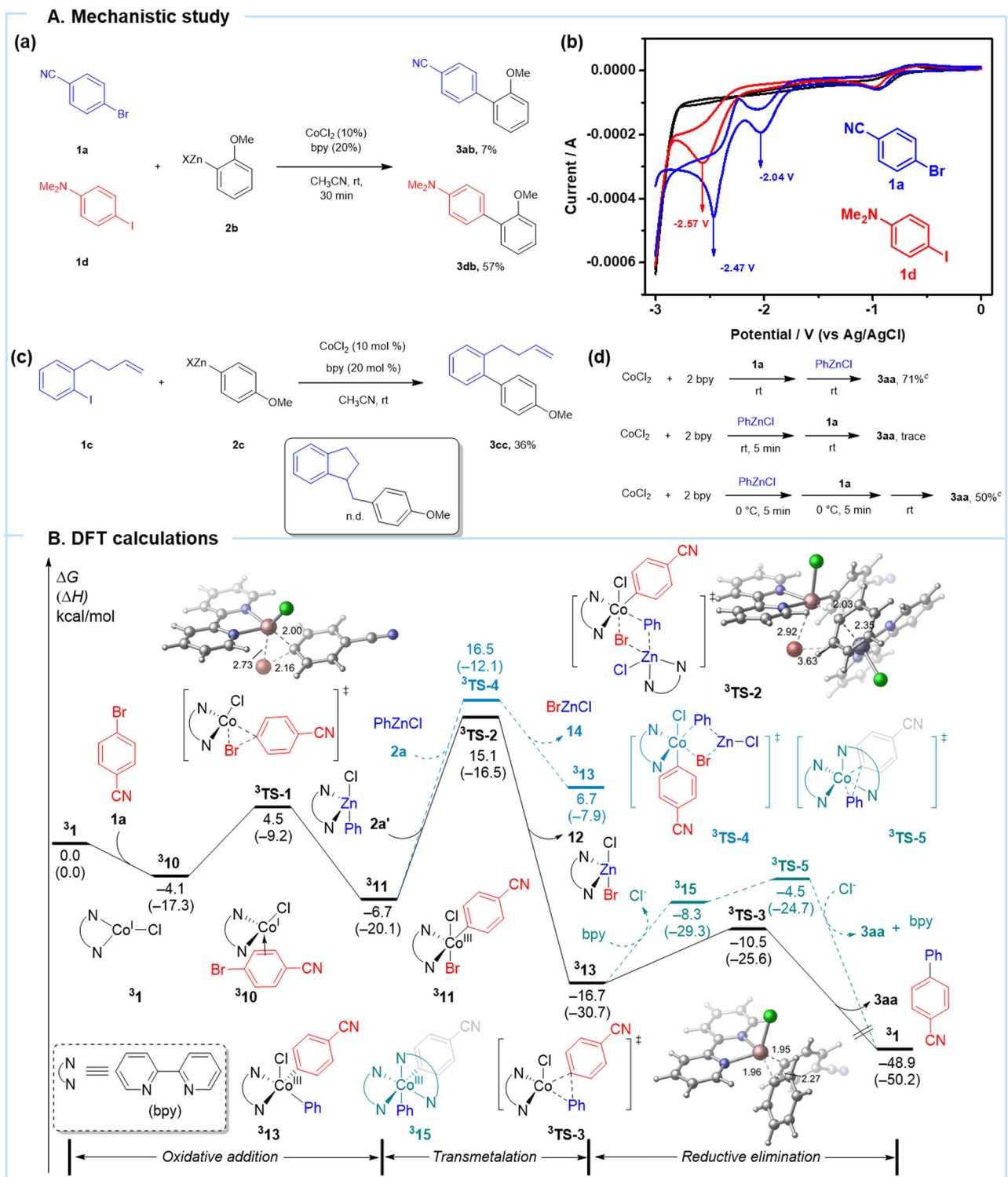

**Fig. 4 | Experimental studies and DFT calculation. A** Mechanistic studies. a Substrate competition study for co-catalyzed Negishi-type coupling in CH₃CN, **1a** (0.2 mmol), **1d** (0.2 mmol), **2b** (0.2 mmol). b Cyclic voltammetry (CV) data using tetrabutylammonium hexafluorophosphate as electrolyte. (Blue line represents **1a**, the red line is **1d**). c Radical clock experiment for Co-catalyzed Negishi-type coupling in CH₃CN, **1c** (0.2 mmol), **2c** (0.3 mmol). d Add a sequence of electrophile **1a** (0.2 mmol) and PhZnCl (0.3 mmol) at rt or 0 °C, CoCl₂ (0.02 mmol), bpy (0.04 mmol). ᶜ GC yield using naphthalene as the internal standard. **B** The Gibbs free energy profile of triplet Co(I)-catalyzed cross-coupling reaction. Calculations were performed at the (U)M06/def2-TZVP/SMD(acetonitrile)//B3LYP-D3/def2-SVP level of theory, and bond lengths are shown in Angstroms (Å).

According to the experimental results, a plausible reaction mechanism is presented in Fig. 5B. The reduction of [CoCl₂(bpy)₂] with PhZnCl forms Co(I) species **1** (Fig. 5B-a). The key catalytic intermediate leads to Co₍ₛ₎ and Co(II) by disproportionation in the absence of electrophile[47]. DFT calculations indicate a favored oxidative addition between Co(I) species **³1** and Ar²Br, leading to the formation of Co(III) species **³10**. Thereafter, Co(III) species **³11** undergoes a facile transmetalation and reductive elimination in the case of electron-rich zinc

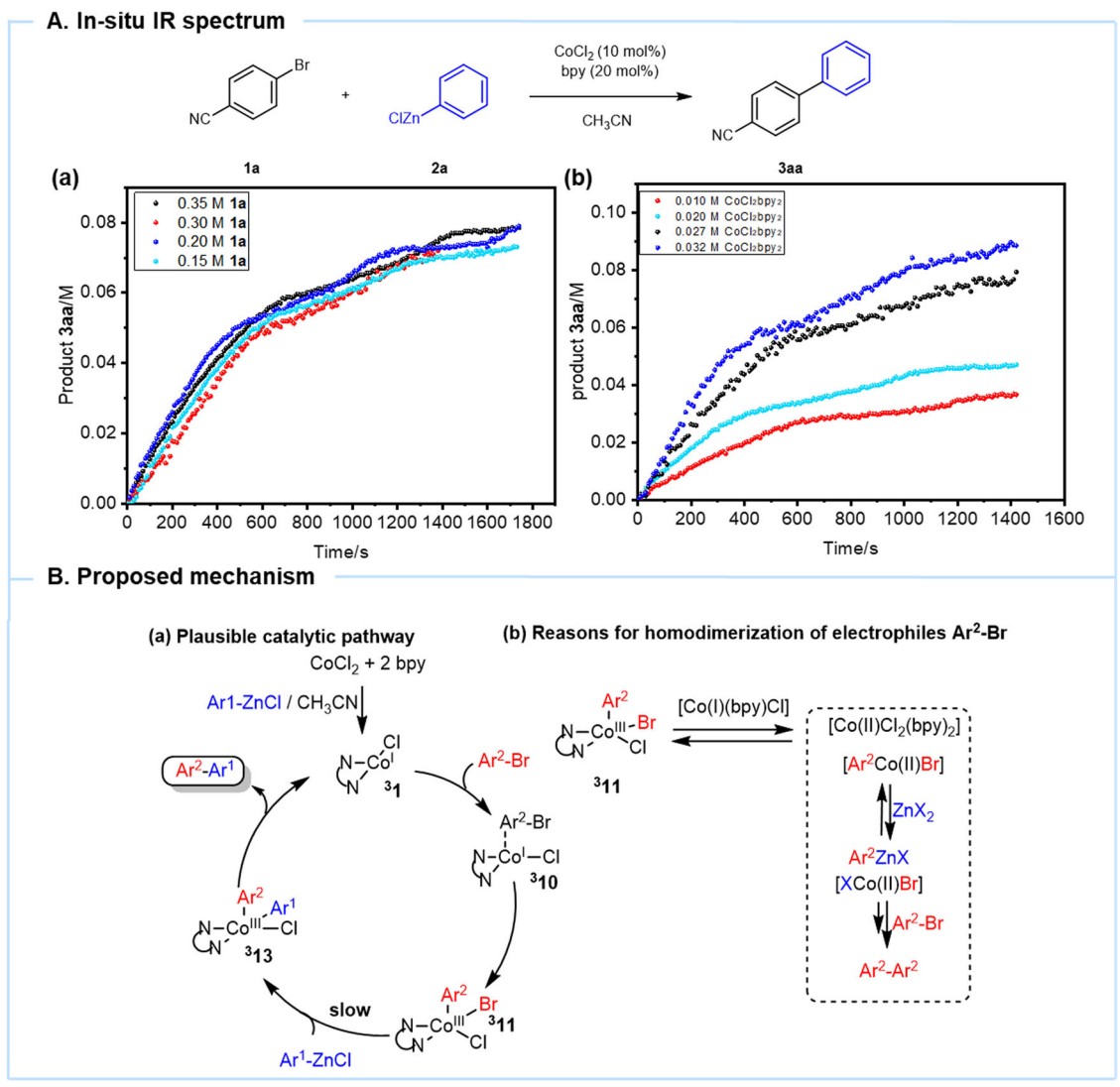

**Fig. 5 | Kinetic study and proposed mechanism. A** In situ IR spectrum. **B** Proposed reaction mechanism.

reagents to afford the desired cross-coupling products. However, with electron-poor nucleophiles and electron-rich electrophiles, the trans-metallation rate could be relatively slow. Therefore, the arylcobalt(III) complex [3]**11** is reduced in solution (Fig. 5B-b), and $Ar^2ZnX$ might be generated, providing the homocoupling product $Ar^2–Ar^2$.

## Scope of substrates

With the mechanism in hand, we began to evaluate the utilization of cobalt-catalyzed Negishi-type cross-coupling. The electronic and steric effects on the coupling were evaluated using various aryl halides (Fig. 6). Electrophiles bearing electron-withdrawing groups efficiently reacted with zinc reagents to afford the desired products in 71–93% yield (3aa–3ib). It is noteworthy that a range of critical functional groups, such as nitriles, esters, and ketones, were tolerated. Meanwhile, dihalo-aryl electrophilic derivatives, such as iodo/bromo and bromo/chloro, aryl derivatives showed high chemo-selectivity, and the products 3jb–3kb were isolated in decent yields. The triaryl product 3lb was generated in 33% yield using 2.5 equiv of the organozinc reagent. Rare examples of electron-rich electrophiles underwent cobalt-catalyzed Negishi-type cross-couplings. The optimized conditions were extended to electrophiles bearing electron-donating groups to afford compounds 3mb–3sb in 53–65% yield. Moreover, aryl derivatives substituted by a vinyl and pyrene group underwent a smooth cross-coupling reaction (3tu–3ub, 62–85% yield).

The substrate scope was extended to various heteroaryl halides (pyrimidine, quinoline, furan, benzothiophene, indole, oxazole, coumarin, pyrazine, pyridine, benzothioxthzole, and purine) to afford the biaryl products 3va–3a3b in 40–89% yields. It is worth noting that heteroaryl chlorides were also reactive enough to efficiently afford the desired cross-coupling products 3a4b–3a8b (34–99% yield). In addition, styryl and phenylethynyl bromide were also reactive in cross-coupling (3a9b–3a10b, 40–80% yield). Furthermore, we examined the reactivity of various functionalized arylzinc reagents in the cobalt-catalyzed Negishi-type cross-couplings. Remarkably, aryl zinc reagents bearing electron-donating and electron-withdrawing groups underwent cross-coupling to generate products 3ga–3vo in 63–98% yield. The synthetic utility of Co-catalyzed cross-coupling is further demonstrated in acylation (3a11f–3a13f), allylation (3a14c–3a15b)[39,48–52], and benzylation (3a16b), which were accomplished in comparable yields to those obtained with Pd catalysts. The concept of regulating solvents and ligands is expected to be extended to other catalytic cross-coupling reactions[53,54].

In summary, a Co(I)/Co(III) catalytic cycle has been revealed by multiple in-situ spectroscopies. We found that the disproportionation of Co(I) to cobalt(0) deposition can be suppressed by mediating solvent and ligand. Co(0) was detected in THF, while Co(I) was detected in $CH_3CN$ and bipyridine system. The formation of highly active Co(I) species through the reduction of Co(II) has been verified, providing the

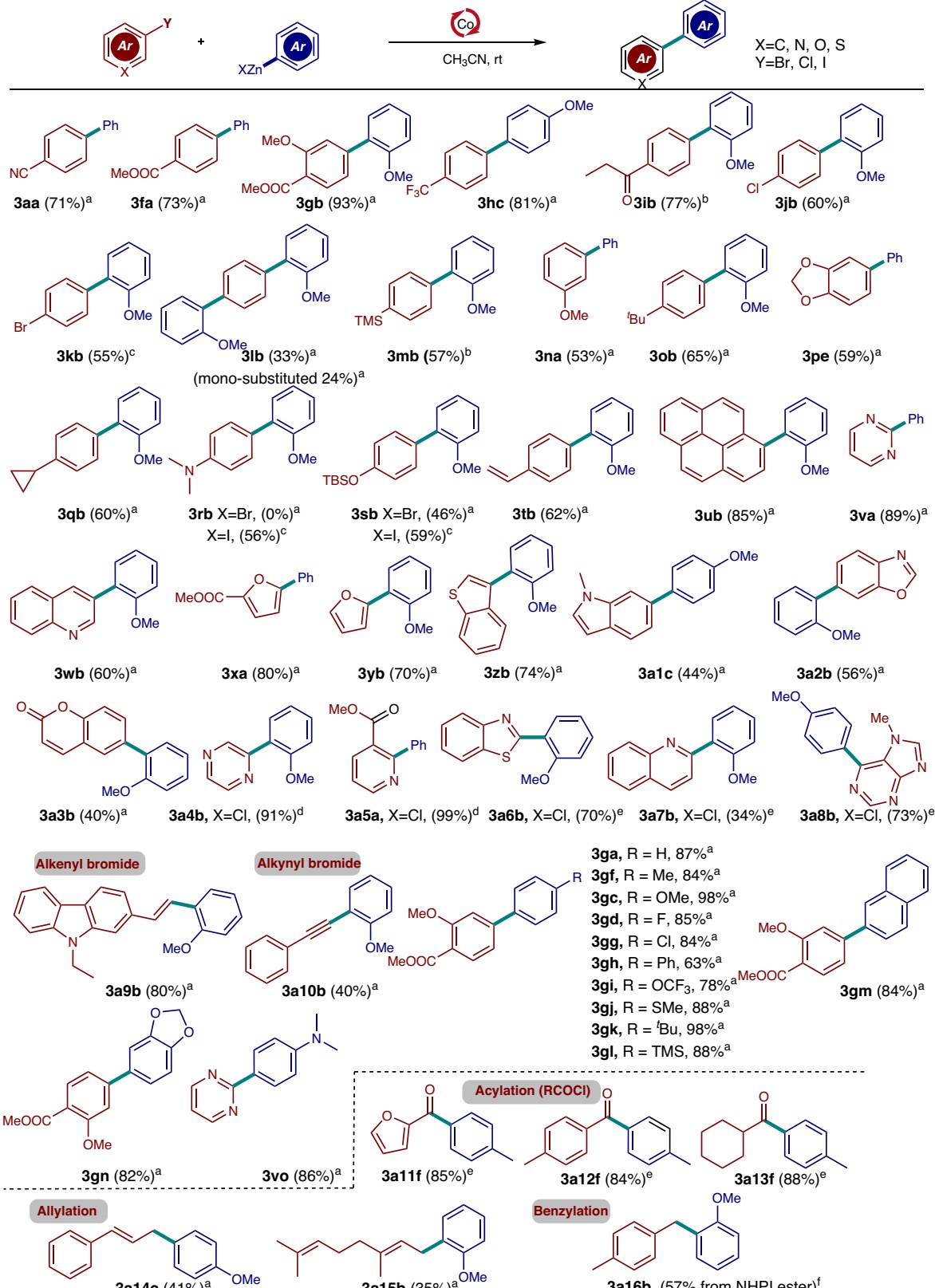

**Fig. 6 | Substrate scopes of Negishi-type cross-couplings[a-d].** [a]Reaction conditions (Method A): bromides (0.2 mmol), zinc reagents (0.3 mmol), CoCl₂ (0.02 mmol), bpy (0.04 mmol), CH₃CN (1 mL). [b]Reaction conditions (Method B): bromides (0.2 mmol), zinc reagents (0.3 mmol), CoCl₂ (0.02 mmol), CH₃CN (1 mL). [c]Reaction conditions (Method A): iodides (0.2 mmol) instead. [d]Reaction conditions (Method B): chlorides (0.2 mmol) instead. [e]Reaction conditions (Method A): chlorides or acyl chlorides (0.2 mmol) instead. [f]Reaction conditions (Method A): N-hydroxyphthalimide (NHPI) ester (0.2 mmol) as electrophile instead[55]. All the reaction times were determined by thin-layer chromatography analysis after the disappearance of the reactants.

direct spectral structure information for Co(I) species. In combination with DFT and in situ IR, it is further clarified that the transmetalation process might be the rate-determining step. Our mechanistic insights guide us to develop an efficient cobalt-catalyzed Negishi-type cross-coupling, which can be extended to acetylation, allylation, and benzylation of aryl zinc derivatives with a wide functional group tolerance. Further application of our mechanistic study for metal deposition in metal catalysis will be a future direction in our laboratory.

## Methods

A 10 mL schlenk tube was equipped with a magnetic stir bar. Electrophiles (0.2 mmol), $CoCl_2$ (0.02 mmol, 2.6 mg), 2,2'-bipyridine (0.04 mmol, 6.3 mg) were added to the schlenk tube. Then MeCN (0.5 mL) was added and stirred for seconds. Corresponding solid organozinc reagents were added, and the remaining MeCN (0.5 mL) was added along the wall of the tube. The reaction was monitored by TLC and GC–MS to identify the reaction time. And the reaction was stopped when electrophiles were fully converted. The reaction was quenched by saturated $NH_4Cl$ aqueous solution and extracted with ethyl acetate three times. The combined organic layer was dried over anhydrous $Na_2SO_4$ and was evaporated in a vacuum. The desired products were obtained in the corresponding yields after purification by flash chromatography on 200–300 mesh silica gel. Full experimental procedures are provided in the Supplemental Information.

## Data availability

All data of this study are available within the article and its Supplementary Information. Crystallographic data for the structures reported in this article have been deposited at the Cambridge Crystallographic Data Center (CCDC) under deposition number CCDC 2040561. A source file displaying the coordinates of the optimized structures is present. Source data are provided in this paper.

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

## Acknowledgements

This work was supported by the National Key R&D Program of China No. 2022YFA1505100, 2021YFA1500100, the National Natural Science Foundation of China (22031008, 21520102003, 21702152, 22271034,), Guangdong Basic and Applied Basic Research Foundation 2023A1515012260, the Fundamental Research Funds for the Central Universities (2042022rc0030, 2042023kf0108, 2042023kf1002). The authors acknowledge the support provided by K.A.CARE-King Abdulaziz University Collaboration Program. XAS study was performed at National Synchrotron Radiation Research Center (44 A in Taiwan Photon Source). The numerical calculations in this paper have been done on the supercomputing system in the Supercomputing Center of Wuhan University. We thank Prof. Oliver Muller from Bergische University Wuppertal for providing JAQ Analyzes QEXAFS software in analysis Quick-XAFS data. We thank Dr. Baosheng Wei for the helpful discussions.

## Author contributions

A. Lei., H. Yi. Y. Chen. conceived the project and designed the experiments. X. Luo., D. Yang., X. He., S. Wang., D. Zhang., J. Xu., C. Pao., J. Chen., J. Lee., H. Cong., Y. Lan., H. Alhumade. performed and analyzed experiments. X. Luo., J. Cossy., R. Bai., Y. Chen., H. Yi.,and A. Lei. wrote the manuscript. All the authors discussed the results and commented on the manuscript.

## Competing interests

The authors declare no competing interests.
