## [Peer Review File · Nature Communications]

REVIEWER COMMENTS

Reviewer #1 (Remarks to the Author):

Review of manuscript titled "Valve Turning towards On-Cycle in Cobalt-Catalyzed Negishi-type Cross-coupling"

The authors Lei et al have explored the impact of solvent (MeCN) and additional supporting ligands (bipy) towards significantly increasing the on-cycle component of cobalt catalyzed Negishi-type cross coupling reaction. The authors have used a combination of spectroscopic techniques and DFT methods to characterize the species involved in as intermediates, both in the on-cycle and off-cycle pathways. Because Co(I) and Co(III) are S=0 species, the authors have relied on XAS methods to characterize the intermediates. As a reviewer with strong expertise in x-ray spectroscopy methods, this manuscript presents clear evidence for the formation of the Co(I) species that goes on to form the Co(III) species in the on-cycle case and the rapid formation of Co(0) species in the off-cycle case. The results are novel and this manuscript should be considered for publication in Nature Communications.

Some specific comments that the authors should address before publication:

While Figure 2A(III) does show some similarities between the FT of Co(0) foil and the off-cycle deposited metallic cobalt, there are significant differences to higher R. This indicates the reduced Co form, while displaying Co-Co distances is not similar to a Co foil spectrum and is likely a different Co(0) structure.

The spectra shown in Fig2B(III) and (IV) indicate that the species is quite different from CoCl₂ and that we are now dealing with a completely different molecular catalyst that CoCl₂. Thus the authors need to modify scheme 1(right hand side) to show that L is already present in the Co(II)X₂ starting species. The way it is presented right now indicates that the ligand is added after the reduction of Co(II) to Co(I) is achieved. This is also true when we look at the synthesis in page S3 in the SI, first the CoCl₂ and the bipy solution are stirred with the electrophile and then the reducing agent is added. Thus, the Co(bpy)₂Cl₂ species is formed first and then reduced.

Based on this above comment, this reviewer is somewhat confused by Figure 1. If the organoZn compound is the reductant then is it clear that the first addition is by Ar₂ and the second addition is Ar₁. If this is not known then it should be mentioned in the mechanism.

The reviewer also suggests that the change of title or abstract to include that Co(II)(bpy)₂Cl₂ is the true, improved catalyst over CoCl₂.

The authors then apply XAS methods in figure 3 to follow the mechanism of cross coupling. The analysis presented here needs clarification. The blue spectrum representing $\text{CoCl}_2\text{bpy}_2$, and PhZnCl (10 equiv.) in CH_3CN does not appear to be Co(I) . It appears to be significantly converted to a Co(III) compound. Is this possible if the Co(II) is first reduced by the 10 equiv excess PhZnCl and then to Co(III) by forming a Co(III)Ph complex? The XAS spectrum does not resemble a Co(I) and indicates a species which may have some Co(I) but mostly Co(III) . This is also consistent with the FT data presented in 3b. The blue and purple spectra are nearly identical with the purple one having slightly higher intensity. This means that it is likely that the Co(III) is formed. Having said this, the EXAFS are of rather poor quality (upto $k=10$) and high noise in the blue spectrum. The authors also do not show the EXAFS fits to the purple spectrum, which is expected to be very similar to the blue spectrum suggesting that the structures are similar. The Co(III) nature of the blue and purple compounds are also indicated by the high intensity of the pre-edge region.

The authors in line 152 say that they have done ORCA calculations. There is no methods section to really understand the details of the calculations and the pre-edge moving 17.2 eV makes no sense. Typical pre-edge shifts will be between 1-2 eV's with oxidation state. This entire figure and statement should be removed.

The authors should redo the EXAFS fits of the blue spectrum considering the formation of a Co(III) species. Given the EXAFS data quality, there could still be two bipy ligands present.

The authors say (line 182-194) that the reaction of Co pre catalyst with the Zn complex could lead to disproportionation without the electrophile. However this goes against the blue spectrum they have presented in Figure 3 which does not show the formation of a Co(0) complex.

Because of these concerns, this reviewer suggests that the authors reevaluate the mechanism they have presented. Additionally there are many grammar mistakes that should be corrected.

Reviewer #2 (Remarks to the Author):

Noble transition metal-catalyzed cross-coupling reactions are quite important. Compared to noble transition metals, earth-abundant transition metals play an increasing role, however, the mechanistic studies are extremely challenging. This manuscript described a valve turning towards on-cycle mechanism for cobalt-catalyzed Negishi-type cross-coupling. To discover the details of this transformation, Quick-X-Ray Absorption Fine Structure (Q-XAFS) spectroscopy, Electron Paramagnetic Resonance (EPR), IR allied with DFT calculations have been used. The scope of

substrate is quite broad. The SI is well prepared. The manuscript could be considered after addressing the following comments.

Comments:

For the SI, the amount of each product should be listed. The original literatures should be cited for the known compounds.

The authors seem to oversell the concept of metal deposition. It is common sense in metal-catalyzed reactions and has been studied in many different cases. The first two sentences in the abstract do not help the whole manuscript and should be deleted.

Reviewer #4 (Remarks to the Author):

The manuscript by Lei and co-authors reported a mechanistic study of a cobalt-catalyzed Negishi-type cross-coupling. The authors tried to elucidate a catalytic cycle using various spectroscopic studies. This study is very difficult to understand for a non specialist certainly because it is not well written. However it would be interesting to complete this study with an X-Ray structure of the proposed CoI to prove its structure. For the catalytic studies, arylzinc species were formed from Grignard reagents implying the absence of reactive functional groups on the ArZnX. I also wonder why the majority of cross-couplings was carried out with a methoxy in ortho position on the arylzinc species. The formation of arylzinc species by Gosmini's method in acetonitrile would be of great interest to have many reactive functional groups on the aromatic moiety and to avoid the removal of THF. Moreover, the Co-catalyzed acylation was already reported without bipyridine (Tetrahedron 2003, JOC 2004), the cross-coupling with heteroaryl compound too (JOC 2009) and also the benzylation (Chem Comm 2008).

Then, in my opinion, although this manuscript certainly presents an interesting study of a CoI/CoIII catalytic cycle by various in situ spectroscopies, all the cross-coupling reactions are not new except in the case of Negishi cross-coupling with aryl halides. This manuscript is not suitable for publication in Nature Communication in this state.

Reviewer #5 (Remarks to the Author):

In this manuscript, Lei and co-workers demonstrated a Cobalt-Catalyzed Negishi-type Cross-coupling reaction for valve turning towards an on-cycle mechanism. Good yields and excellent substrate scopes were reported. Moreover, solid evidence of the full catalytic cycle is achieved by using multiple spectroscopic studies, such as Quick-X-Ray Absorption Fine Structure (Q-XAFS) spectroscopy, Electron Paramagnetic Resonance (EPR), IR and DFT calculations. These comprehensive mechanistic insights establish the structural information of the catalytic active Co(I) species and demonstrate the whole catalytic Co(I)/Co(III) cycle. Mechanistic study reveals an effective way to improve the Negishi-type cross-coupling systems by altering solvents and ligands and should be of interest to synthetic chemists. Overall, this work is significant and well done, it is novel enough to be considered for the journal Nature Communications. This reviewer would like to suggest its acceptance with Minor modifications:

1. Page 5, line 2: There is an extra dot before the comma. In page 6, line 10, space need to be required between the unit and the number
2. Page 8, lines 22 and 24: Whether the description should be “intermediate 313.....” rather than 13T?
3. Is there any other spectroscopy studies you have tried apart from these methods mentioned in the paper, such as XPS and UV?
4. It is very tempting to test substituted bipyridines as well as TMEDA as additives. Are there other ligands you have tried to further improve the efficiency and generality of the catalytic system? And is there any limited scope you have tried in this condition?
5. In Scheme S1 of supporting information, there is a blank space between mol and %. The correct form should be 10 mol%. And one “equiv.” loses a dot in page 4, SI.
6. The spatial configuration of structure 13 and structure 15 in Figure 4B is not clear, please double-check in the DFT part.

Response Letter

Reviewer #1 (Remarks to the Author):

The authors Lei et al have explored the impact of solvent (MeCN) and additional supporting ligands (bipy) towards significantly increasing the on-cycle component of cobalt catalyzed Negishi-type cross coupling reaction. The authors have used a combination of spectroscopic techniques and DFT methods to characterize the species involved in as intermediates, both in the on-cycle and off-cycle pathways. Because Co(I) and Co(III) are S=0 species, the authors have relied on XAS methods to characterize the intermediates. As a reviewer with strong expertise in x-ray spectroscopy methods, this manuscript presents clear evidence for the formation of the Co(I) species that goes on to form the Co(III) species in the on-cycle case and the rapid formation of Co(0) species in the off-cycle case. The results are novel and this manuscript should be considered for publication in Nature Communications.

Our response: Thanks very much for the appreciation of our work.

Some specific comments that the authors should address before publication:

1. While Figure 2A(III) does show some similarities between the FT of Co(0) foil and the off-cycle deposited metallic cobalt, there are significant differences to higher R. This indicates the reduced Co form, while displaying Co-Co distances is not similar to a Co foil spectrum and is likely a different Co(0) structure.

Our response: Thanks very much for the comments. Indeed, the Co(0) formed from $\text{CoBr}_2 + 5 \text{PhZnCl}$ is a different Co(0) from Co foil. It is assigned as Co(0) because of the similarity of edge position and no pre-edge feature in XANES. The FT EXAFS shows the similarity of the peak position but distinct amplitude also different higher R features. This is indicating the off-cycle Co(0) is a Co nanoparticle instead of metallic cobalt. We add EXAFS fitting results here to prove that formed Co(0) has a lower coordination number and shorter Co-Co bond length on average, which all indicates the formation of small Co(0) nanoparticles. We add this data in the revised SI, Figure S3.

	Path	amp	CN	R	DW	R-factor
Co foil	Co-Co	0.79	12	2.495	0.0064	0.00124
CoBr ₂ + 5PhZnCl	Co-Co	0.79	7 (1)	2.442	0.0150	0.01234

2. The spectra shown in Fig2B(III) and (IV) indicate that the species is quite different from CoCl₂ and that we are now dealing with a completely different molecular catalyst that CoCl₂. Thus the authors need to modify scheme 1(right hand side) to show that L is already present in the Co(II)X₂ starting species. The way it is presented right now indicates that the ligand is added after the reduction of Co(II) to Co(I) is achieved. This is also true when we look at the synthesis in page S3 in the SI, first the CoCl₂ and the bpy solution are stirred with the electrophile and then the reducing agent is added. Thus, the Co(bpy)₂Cl₂ species is formed first and then reduced.

Based on this above comment, this reviewer is somewhat confused by Figure 1. If the organozinc compound is the reductant then it is clear that the first addition is by Ar² and the second addition is Ar¹. If this is not known then it should be mentioned in the mechanism.

Our response: Thanks very much for the comments. In our reaction, some substrates with electron withdraw groups can be tolerant without ligand. So in the previous fig1(b), Co(I) can also be formed without bpy, while under disproportionation of Co(I) to cobalt(0) deposition. Based on your suggestions, we have redrawn the fig1(b). The first step of the organozinc reagent Ar²ZnCl is to provide electrons for Co(bpy)₂Cl₂. Because the reductive species we have fitted as Co(bpy)Cl do not contain Ar². Later on, Ar²ZnCl participates in the transmetalation process after the oxidative addition of Ar¹X.

3. The reviewer also suggests that the change of title or abstract to include that Co(II)(bpy)₂Cl₂ is the true, improved catalyst over CoCl₂.

Our response: Thanks very much for the comments. We use “cobalt” in title because not only $\text{CoCl}_2\text{bpy}_2$ but also CoCl_2 can catalyze Negishi-type cross-coupling, although $\text{CoCl}_2\text{bpy}_2$ can be more effective than CoCl_2 . We have modified abstract and highlight $\text{CoCl}_2\text{bpy}_2$.

4. The authors then apply XAS methods in figure 3 to follow the mechanism of cross coupling. The analysis presented here needs clarification. The blue spectrum representing $\text{CoCl}_2\text{bpy}_2$, and PhZnCl (10 equiv.) in CH_3CN does not appear to be Co(I) . It appears to be significantly converted to a Co(III) compound. Is this possible if the Co(II) is first reduced by the 10 equiv excess PhZnCl and then to Co(III) by forming a Co(III)Ph complex? The XAS spectrum does not resemble a Co(I) and indicates a species which may have some Co(I) but mostly Co(III) . This is also consistent with the FT data presented in 3b. The blue and purple spectra are nearly identical with the purple one having slightly higher intensity. This means that it is likely that the Co(III) is formed. Having said this, the EXAFS are of rather poor quality (upto $k=10$) and high noise in the blue spectrum. The authors also do not show the EXAFS fits to the purple spectrum, which is expected to be very similar to the blue spectrum suggesting that the structures are similar. The Co(III) nature of the blue and purple compounds are also indicated by the high intensity of the pre-edge region.

Our response: Thanks very much for the comments. EXAFS does not tell the oxidation state. One cannot compare FT similarity to determine the oxidation state of Co. the pre-edge calculation does. We are confident to say Co(I) instead of Co(III) is formed because PhZnCl is a strong reductant and was added with an excess amount in mechanism studies. After adding $p\text{-CNPhBr}$, the EXAFS signal is of worse quality which makes its EXAFS data hard to fit. The cobalt structure after adding $p\text{-CNPhBr}$ is more likely to be a complex mixture that has some Co(I) remaining and newly formed higher oxidation state Co based on the edge shifts. A clear structure is hard to determine from XAFS. But Co(III) was proposed by the spin quantification vanishing which resembles Co(III) formation instead of Co(II) .

5. The authors in line 152 say that they have done ORCA calculations. There is no methods section to really understand the details of the calculations and the pre-edge moving 17.2 eV makes no sense. Typical pre-edge shifts will be between 1-2 eV's with oxidation state. This entire figure and statement should be removed.

Our response: Thanks very much for the comments. We add ORCA calculation details in the

article. 17.2 eV is the shifts applied to the calculated spectrum to match the experimental data. It is not the energy difference between two calculated spectra. It is common to have the calculated spectrum over ten eV away from the experimental edge. We are comparing the relative difference instead of the absolute value of the pre-edge. We add the methods section for ORCA TD-DFT calculation accordingly.

6. The authors should redo the EXAFS fits of the blue spectrum considering the formation of a Co(III) species. Given the EXAFS data quality, there could still be two bipy ligands present.

Our response: Thanks very much for the comments. We did not do Co(III) EXAFS fits. Because the oxidation state can't be Co(III) under such a reducing condition. EXAFS fits can't be used to determine oxidation states. But we have tried to add Co(I) fits with two bipy. We re-fit the $\text{CoCl}_2\text{bpy}_2 + 10\text{PhZnCl}$ data, considering still having two bpy ligand (shown in green in the above figure and table). The R-factor get slightly better, and the low R parts were better fitted. However, the Debye-Waller factor of the first shell Co-N is incredibly large, which makes the fitting unlikely to be trusted. Pre-edge TD-DFT calculation also supports CobpyCl better than Co with two bpy ligands. We will claim that the reduced Co may be a mixture of CobpyCl and Cobpy₂Cl but the oxidation state can't be Co(III) under such a reducing condition.

More EXAFS Fitting results

Fitting model	Path	CN	σ^2	ΔE (eV)	R (Å)	R-factor
	Co-N	4	0.02360		2.071	
Cobpy ₂ Cl	Co-Cl	1	0.00456	1.0	2.272	0.00403
	Co-C	4	0.01417		2.917	

	Co-N	2	0.00868		2.070	
CobpyCl	Co-Cl	1	0.00758	2.5	2.295	0.00489
	Co-C	2	0.00266		2.931	

Pre-edge TD-DFT calculations

7. The authors say (line 182-194) that the reaction of Co pre catalyst with the Zn complex could lead to disproportionation without the electrophile. However this goes against the blue spectrum they have presented in Figure 3 which does not show the formation of a Co(0) complex.

Our response: Thanks very much for the comments. We prepared and measured the spectrum in low temperatures so that may prevent the disproportionation. In Fig 4Ad, no product was obtained by the sequence of $\text{CoCl}_2(\text{bpy})_2$, PhZnCl and electrophile **1a**. Nevertheless, a moderate yield was observed by premixing $\text{CoCl}_2(\text{bpy})_2$ and PhZnCl in CH_3CN at $0\text{ }^\circ\text{C}$. Then electrophile **1a** was added at $0\text{ }^\circ\text{C}$ for 5 min and stirred at rt. We reasoned that the active Co(I) species were generated under standard conditions and can be stabilized at low temperatures.

8. Because of these concerns, this reviewer suggests that the authors reevaluate the mechanism they have presented. Additionally there are many grammar mistakes that should be corrected.

Our response: Thanks very much for your comments. We have double-checked the whole manuscript and revised it accordingly.

Reviewer #2 (Remarks to the Author):

Noble transition metal-catalyzed cross-coupling reactions are quite important. Compared to noble transition metals, earth-abundant transition metals play an increasing role, however, the mechanistic studies are extremely challenging. This manuscript described a valve turning towards on-cycle mechanism for cobalt-catalyzed Negishi-type cross-coupling. To discover the details of this transformation, Quick-X-Ray Absorption Fine Structure (Q-XAFS) spectroscopy, Electron Paramagnetic Resonance (EPR), IR allied with DFT calculations have been used. The scope of substrate is quite broad. The SI is well prepared. The manuscript could be considered after addressing the following comments.

Our response: Thanks very much for the appreciation of our work.

Comments:

1. For the SI, the amount of each product should be listed. The original literatures should be cited for the known compounds.

Our response: Amount of products and original literatures are added in SI.

2. The authors seem to oversell the concept of metal deposition. It is common sense in metal-catalyzed reactions and has been studied in many different cases. The first two sentences in the abstract do not help the whole manuscript and should be deleted.

Our response: Thanks for the suggestions. According to the reviewer's suggestion, we have changed our abstract in the revised manuscript.

Reviewer #4 (Remarks to the Author):

The manuscript by Lei and co-authors reported a mechanistic study of a cobalt-catalyzed Negishi-type cross-coupling. The authors tried to elucidate a catalytic cycle using various spectroscopic studies. This study is very difficult to understand for a non specialist certainly because it is not well written. However it would be interesting to complete this study with an X-Ray structure of the proposed CoI to prove its structure. For the catalytic studies, arylzinc species were formed from Grignard reagents implying the absence of reactive functional groups on the ArZnX. I also wonder why the majority of cross-couplings was carried out with a methoxy in ortho position on the arylzinc species. The formation of arylzinc species by Gosmini's method in acetonitrile would be of great interest to have many reactive functional groups on the aromatic moiety and to avoid the removal of THF. Moreover, the Co-catalyzed acylation was already reported without bipyridine (Tetrahedron 2003, JOC 2004), the cross-coupling with heteroaryl compound too (JOC 2009) and also the benzylation (Chem Comm 2008).

Our response: Thanks for the suggestions. We have added these references in the corresponding locations. We used methoxy in *ortho* position on the arylzinc species because the homo-coupling of organozinc reagent will rise when using para or meta reagents. It is true that the excellent work reported by Gosmini can avoid the removal of THF. However, we focus on the mechanistic study. At the same time, we provided a general method not only for electron-withdraw scopes but also for electron-donating groups.

Then, in my opinion, although this manuscript certainly presents an interesting study of a CoI/CoIII catalytic cycle by various in situ spectroscopies, all the cross-coupling reactions are not new except in the case of Negishi cross-coupling with aryl halides. This manuscript is not suitable for publication in Nature Communication in this state.

Our response: Thanks very much for your comments. In our manuscript, the structure and reactivity of Co(I) species in Co-catalyzed Negishi cross-coupling efficiently have been well-defined and identified. Although Negishi coupling was reported before, these conditions were limited to highly active electron-deficient electrophiles. The reaction mechanism is still unclear. X-ray absorption fine structure spectroscopy (XAFS), electron paramagnetic resonance (EPR), and synthesis confirm a Co(I)/Co(III) catalytic cycle including structural information of the catalytic

intermediates. The resulting Co(I) species are stabilized by ligand and solvent, preventing the disproportionation. The mechanistic studies provide reaction conditions with a comprehensive substrate scope that can further extend to acetylation, allylation, and benzylation. Overall, our work provides a basic mechanistic perspective for designing cobalt-catalyzed cross-coupling reactions.

Reviewer #5 (Remarks to the Author):

In this manuscript, Lei and co-workers demonstrated a Cobalt-Catalyzed Negishi-type Cross-coupling reaction for valve turning towards an on-cycle mechanism. Good yields and excellent substrate scopes were reported. Moreover, solid evidence of the full catalytic cycle is achieved by using multiple spectroscopic studies, such as Quick-X-Ray Absorption Fine Structure (Q-XAFS) spectroscopy, Electron Paramagnetic Resonance (EPR), IR and DFT calculations. These comprehensive mechanistic insights establish the structural information of the catalytic active Co(I) species and demonstrate the whole catalytic Co(I)/Co(III) cycle. Mechanistic study reveals an effective way to improve the Negishi-type cross-coupling systems by altering solvents and ligands and should be of interest to synthetic chemists. Overall, this work is significant and well done, it is novel enough to be considered for the journal Nature Communications. This reviewer would like to suggest its acceptance with Minor modifications:

1. Page 5, line 2: There is an extra dot before the comma. In page 6, line 10, space need to be required between the unit and the number

Our response: We are thankful for reviewer's comments, which help us improve our work. We have carefully examined formatting issues and made the corresponding changes.

2. Page 8, lines 22 and 24: Whether the description should be "intermediate 313" rather than 13T ?

Our response: We are thankful for reviewer's comments, we have made corresponding modifications in the revised manuscript.

3. Is there any other spectroscopy studies you have tried apart from these methods mentioned in the paper, such as XPS and UV?

Our response: XPS and UV studies have been tried. However, we cannot get more information. We want to prove Co(0) species in CoCl₂ and PhZnCl system. But no Co(0) signal was not detected, because the reductive Co(0) species is easily oxidized by O₂ during the test.

As for UV studies, we have tried to monitor the change of cobalt valence in the reaction. The UV-Vis was tested under room temperature initially from 190 nm to 1100 nm, there was no difference

after the addition of PhZnCl to CoCl₂bpy₂. So we reduced the test temperature to 0 °C and improved the reagent concentrations. [UV-Vis experimental conditions: CoCl₂bpy₂ (10⁻³ M); CoCl₂ (10⁻³ M); bpy (2*10⁻³ M); PhZnCl (10⁻² M) and pCNArBr (10⁻² M)].

According to the literature report, the absorption bands of CoCl₂ in CH₃CN (10⁻³ M) in the range of 200-300 nm may assign to ligand-based $\pi \rightarrow \pi^*$ transitions. And the absorption bands of 500-735 nm may correspond to the excitation into a Co-Co σ^* manifold. The experimental results suggested that CoCl₂ may be dimerized in CH₃CN. (*Inorg. Chem.* **2020**, *59*, 4200–4214).

The CoCl₂bpy₂ does not have the same absorption peak in range of 500-735 nm, which reveals the complexation of bpy and CoCl₂ instead of dimerized complex.

Fig 1

It is very difficult to obtain key catalytic intermediates through synthesis as a standard for comparison of UV-Vis spectrum, due to the poor stability of the reaction intermediates. The reduction process was monitored by using UV-Vis spectra as shown in Fig. 1g. When we added PhZnCl to CoCl₂bpy₂, there was a new wavelength at 542 nm in the initial stage of reaction, which means the interaction between CoCl₂bpy₂ and PhZnCl. It is noteworthy that the spectroscopy

changes upon the reaction mixture was stirred and two peaks (589 nm and 690 nm) remain after 1 h stirring. It has been reported that Co(I) species is tentatively associated with intraligand charge transfer (ILCT) and metal-to-ligand charge transfer (MLCT) bands at 572 nm and 685 nm. (*Chem. Sci.*, **2016**, *7*, 3264–3278.) We speculate that Co(I) may be formed under the conditions of precatalyst activation process. However, not much literature support this phenomenon. The system is relatively complicated. We are not confirmed.

4. It is very tempting to test substituted bipyridines as well as TMEDA as additives. Are there other ligands you have tried to further improve the efficiency and generality of the catalytic system? And is there any limited scope you have tried in this condition?

Our response: We have tried different ligands including 1,10-phenanthrolines as shown in the scheme. However, the reaction were not further improved comparing with bpy (Fig 2). As for scope with active hydrogen (Fig 3), the reactivity is poor due to the strong alkaline of organozinc reagents.

Fig 2

Fig 3

5. In Scheme S1 of supporting information, there is a blank space between mol and %. The correct

form should be 10 mol%. And one “equiv.” loses a dot in page 4, SI.

Our response: We have modified SI.

6. The spatial configuration of structure 13 and structure 15 in Figure 4B is not clear, please double-check in the DFT part.

Our response: Thanks for the suggestions, which help us improve our work. We have made corresponding modifications in the revised manuscript.

REVIEWERS' COMMENTS

Reviewer #1 (Remarks to the Author):

The authors have replied to my concerns satisfactorily, there are 2 remaining concerns that they should address before publication.

Minor

1) The authors say (line 150) : In addition, ORCA pre-edge calculation showed that the pre-energy of Co(I) species 1 shifted 17.2 eV from the initial [CoCl₂(bpy)₂] catalyst (Figure 3g). This statement is incorrect. It is talking about a 17.2 eV shift between the 2 species and not talking about the relative shift between experimental and. Calculated spectra. This should be fixed.

2) The 17.2 value is unlikely to be correct and this reviewer thinks this is a typo. It should be closer to 170 eV. In the reference that the reviewer gives "J. Phys. Chem. 112, 12936–12943 (2008)", the shift is ~171 eV.

Authors need to think about this:

3) Figure 3(a): 10PhZCl should be 10PhZnCl.

Second while this reviewer completely agrees that the spectrum in the excess condition product should be Co(I). The authors data unfortunately do not agree with the story that they are telling:

This is the authors response:

Authors: Thanks very much for the comments. EXAFS does not tell the oxidation state. One cannot compare FT similarity to determine the oxidation state of Co.

Indeed, the EXAFS and FT cannot give oxidation state. But in your case the FT are nearly IDENTICAL to each other. The authors are saying that the Co(I) and a Co(III) species have nearly identical structures with very similar bond distances. This does not appear to be true.

Authors: The pre-edge calculation does.

It is not the pre-edge calculation that shows whether it is a Co(I) or a Co(III) species, but it is the energy positions of the pre-edge and the rising-edge features. Note that your rising-edge and pre-edges are again, very similar for the blue and purple lines (CoI and CoIII species).

Authors: We are confident to say Co(I) instead of Co(III) is formed because PhZnCl is a strong reductant and was added with an excess amount in mechanism studies.

The reviewer does not doubt that in the presence of excess reductant the product should be Co(I). But the spectra are not that of a typical Co(I) species. They resemble the Co(III) species presented here (purple line). The authors need to explain this, without the hand wavy explanation in the first round of reviews.

It is possible that since the Co(I) is a very different species from the Co(II) and has very short Co-N bonds (as shown from the EXAFS fits), It is therefore also likely that there is a significant amount of back-bonding character in the Co(I) with the bipy ligand and the post-edge of Co(I) blue line is shifted to higher energy than Co(II). This should be included in the manuscript.

In either case, the authors have 2 choices:

- 1) Either the purple line is not fully converted in the XAS spectrum and has some significant amount of Co(I) and is actually a mixture of say 40%Co(III) and 60% Co(I).
- 2) Or for some reason, the spectra of Co(I) and Co(III) are very similar and the authors have to justify this.

Without this explanation the reader will not be able to trust the datasets presented in this manuscript.

Reviewer #5 (Remarks to the Author):

The revisions have been carefully done.

The manuscript is now ready for publication.

Response Letter

Reviewer #1 (Remarks to the Author):

The authors have replied to my concerns satisfactorily, there are 2 remaining concerns that they should address before publication.

Minor

1) The authors say (line 150) : In addition, ORCA pre-edge calculation showed that the pre-energy of Co(I) species 1 shifted 17.2 eV from the initial [CoCl₂(bpy)₂] catalyst (Figure 3g). This statement is incorrect. It is talking about a 17.2 eV shift between the 2 species and not talking about the relative shift between experimental and. Calculated spectra. This should be fixed.

Our response: Thanks for the reviewer's comment and we apologize for our previous statement by mistake. The statement in the manuscript was revised: "In addition, ORCA pre-edge TDDFT calculation showed that the pre-edge intensity of Co(I) species 1 matches with the generated Co(I) experimental data well, which is more intense than the pre-edge intensity of [CoCl₂(bpy)₂] catalyst as Co(II) species (Figure 3g)." We added our ORCA calculation details in the SI.

2) The 17.2 value is unlikely to be correct and this reviewer thinks this is a typo. It should be closer to 170 eV. In the reference that the reviewer gives "J. Phys. Chem. 112, 12936–12943 (2008)", the shift is ~171 eV.

Our response: 17.2 eV is not a typo when relativistic effects theory ZORA calculation was performed. Without ZORA, 165 eV absolute energy shift was identified with similar pre-edge shape and intensities.

Authors need to think about this:

3) Figure 3(a): 10PhZCl should be 10PhZnCl.

Our response: We have changed to 10PhZnCl.

Second while this reviewer completely agrees that the spectrum in the excess condition product should be Co(I). The authors data unfortunately do not agree with the story that they are telling:

This is the authors response:

Authors: Thanks very much for the comments. EXAFS does not tell the oxidation state. One cannot compare FT similarity to determine the oxidation state of Co.

Indeed, the EXAFS and FT cannot give oxidation state. But in your case the FT are nearly IDENTICAL to each other. The authors are saying that the Co(I) and a Co(III) species have nearly identical structures with very similar bond distances. This does not appear to be true.

Our response: Thanks for the reviewer's comments and concerns on the Co(III) data. We took the reviewer's criticize that the Co(III) data in figure 3a & 3b is not a fully oxidized Co(III). And also we do not want to mislead the reader to think that the purple line in Figure 3a & 3b is Co(III). We believe also that it is a mixture that may contain Co(I), Co(III) and even Co(II). We draw the conclusion that Co(I) + p-CNPhBr yields Co(III) based on our EPR study that we observed spin vanishing (Figure S2). The only

evidence that matches with this finding in XAFS is the main edge energy blue shift. In our previous response, we also mentioned that “The cobalt structure after adding p-CNPhBr is more likely to be a complex mixture which has some Co(I) remaining and newly formed higher oxidation state Co based on the edge shifts. A clear structure is hard to determine from XAFS. But Co(III) was proposed by the spin quantification vanishing which resembles Co(III) formation instead of Co(II).” Thus, in order to make it more clear and not to mislead the reader, we took out the purple line from Figure 3a and 3b. To emphasis our main message we would like to send our to our readers: Co(I) as the active intermediate can be generated under MeCN, instead, Co(0) is the major product of Co(II) + PhZnCl in THF. Further studies, such as in-situ quick-XAFS is on-going in confirming this process.

Authors: The pre-edge calculation does.

It is not the pre-edge calculation that shows whether it is a Co(I) or a Co(III) species, but it is the energy positions of the pre-edge and the rising-edge features. Note that your rising-edge and pre-edges are again, very similar for the blue and purple lines (CoI and CoIII species).

Our response: Thanks for the reviewer’s comments and concerns on the Co(III) data. As the reason we explained above, we took out the purple line from Figure 3a and 3b. To emphasis our main message we would like to send our to our readers: Co(I) as the active intermediate can be generated under MeCN, instead, Co(0) is the major product of Co(II) + PhZnCl in THF.

Authors: We are confident to say Co(I) instead of Co(III) is formed because PhZnCl is a strong reductant and was added with an excess amount in mechanism studies.

The reviewer does not doubt that in the presence of excess reductant the product should be Co(I). But the spectra are not that of a typical Co(I) species. They resemble the Co(III) species presented here (purple line). The authors need to explain this, without the hand wavy explanation in the first round of reviews.

It is possible that since the Co(I) is a very different species from the Co(II) and has very short Co-N bonds (as shown from the EXAFS fits), It is therefore also likely that there is a significant amount of back-bonding character in the Co(I) with the bipy ligand and the post-edge of Co(I) blue line is shifted to higher energy than Co(II). This should be included in the manuscript.

Our response: Thanks for the reviewer’s suggestions. We added the back-bonding discussion in the manuscript for the Co(I) XANES feature explanation.

In either case, the authors have 2 choices:

- 1) Either the purple line is not fully converted in the XAS spectrum and has some significant amount of Co(I) and is actually a mixture of say 40% Co(III) and 60% Co(I).
- 2) Or for some reason, the spectra of Co(I) and Co(III) are very similar and the authors have to justify this.

Our response: Thanks for the reviewer’s suggestions. We agree with the reviewer that the purple line is a mixture instead of a pure Co(III). But based on our EPR study and other experimental results. We believe the mechanism conclusion we proposed for Co(I) + ArBr → Co(III) is not affected. In order not to mislead the reader, we took out the

purple line in Figure 3a & 3b. A linear combination fraction (LCF) analysis can't be applied here since the active oxidative addition product Co(III) standard is not available. Further efforts in trapping intermediates of this reaction are underway.

Without this explanation the reader will not be able to trust the datasets presented in this manuscript.

Our response: Thanks for the reviewer's comments.